# Evaluation of COVID-19 related knowledge and preparedness in health professionals at selected health facilities in a resource-limited setting in Addis Ababa, Ethiopia

**Zelalem Desalegn**[1]*, **Negussie Deyessa**[2], **Brhanu Teka**[1], **Welelta Shiferaw**[3],
**Meron Yohannes**[4], **Damen Hailemariam**[2], **Adamu Addissie**[2], **Abdulnasir Abagero**[2],
**Mirgissa Kaba**[2], **Workeabeba Abebe**[5], **Alem Abrha**[6], **Berhanu Nega**[7], **Wondimu Ayele**[2],
**Tewodros Haile**[8], **Yirgu Gebrehiwot**[9], **Wondwossen Amogne**[8], **Eva Johanna Kantelhardt**[10],
**Tamrat Abebe**[1]

1 Department of Microbiology, Immunology and Parasitology, College of Health Sciences, School of
Medicine, Addis Ababa University, Addis Ababa, Ethiopia, 2 College of Health Sciences, School of Public
Health, Addis Ababa University, Addis Ababa, Ethiopia, 3 Department of Psychiatry, College of Health
Sciences, School of Medicine, Addis Ababa University, Addis Ababa, Ethiopia, 4 Department of Medical
Laboratory Sciences, College of Health Sciences, Addis Ababa University, Addis Ababa, Ethiopia, 5 Pediatric
and Child Health Department, College of Health Sciences, School of Medicine, Addis Ababa University, Addis
Ababa, Ethiopia, 6 Yekatit-12 Hospital Medical College, Addis Ababa, Ethiopia, 7 Department of Surgery,
College of Health Sciences School of Medicine, Addis Ababa University, Addis Ababa, Ethiopia,
8 Department of Internal Medicine, College of Health Sciences, School of Medicine, Addis Ababa University,
Addis Ababa, Ethiopia, 9 Department of Obstetrics and Gynecology, College of Health Sciences, School of
Medicine, Addis Ababa University, Addis Ababa, Ethiopia, 10 Institute of Medical Epidemiology, Biostatistics
and Informatics, Martin-Luther-University, Halle, Germany

* zelalem.desalegn@aau.edu.et, tzollove@gmail.com

pone.0244050

Sports, MYANMAR

## Abstract

### Background

The World Health Organization has declared that infection with SARS-CoV-2 is a pandemic.
Experiences with SARS in 2003 and SARS-CoV-2 have shown that health professionals
are at higher risk of contracting COVID-19. Hence, it has been recommended that aperiodic
wide-scale assessment of the knowledge and preparedness of health professionals regard-
ing the current COVID-19 pandemic is critical.

### Objectives

This study aimed to assess the knowledge and preparedness of health professionals
regarding COVID-19 among selected hospitals in Addis Ababa, Ethiopia.

### Methods

A facility-based cross-sectional study was conducted from the last week of March to early
April, 2020. Government (n = 6) and private hospitals (n = 4) were included. The front-line
participants with high exposure were proportionally recruited from their departments. The
collected data from a self-administered questionnaire were entered using EpiData and

**Data Availability Statement:** All relevant data are within the manuscript and its Supporting Information files.

**Funding:** ZD and TA received the award from AAU. Grant number: VPRTT/PY-403/2020. The fund was secured from Addis Ababa University. www.aau. edu.et We assure you that the funders had no role in study design, data collection and analysis, decision to publish, or preparation of the manuscript.

**Competing interests:** The authors have declared that no competing interests exist.

analyzed in SPSS software. Both descriptive statistics and inferential statistics (chi-square tests) are presented.

## Results

A total of 1334 health professionals participated in the study. The majority (675, 50.7%) of the participants were female. Of the total, 532 (39.9%) subjects were nurses/midwives, followed by doctors (397, 29.8%) and pharmacists (193, 14.5%). Of these, one-third had received formal training on COVID-19. The mean knowledge score of participants was 16.45 (±4.4). Regarding knowledge about COVID-19, 783 (58.7%), 354 (26.5%), and 196 (14.7%) participants had moderate, good, and poor knowledge, respectively. Lower scores were seen in younger age groups, females, and non-physicians. Two-thirds (63.2%) of the subjects responded that they had been updated by their hospital on COVID-19. Of the total, 1020 (76.5%) participants responded that television, radio, and newspapers were their primary sources of information. Established hospital preparedness measures were confirmed by 43–57% of participants.

## Conclusion

The current study revealed that health professionals in Addis Ababa, Ethiopia, already know important facts but had moderate overall knowledge about the COVID-19 pandemic. There were unmet needs in younger age groups, non-physicians, and females. Half of the respondents mentioning inadequate preparedness of their hospitals point to the need for more global solidarity, especially concerning the shortage of consumables and lack of equipment.

## Introduction

The rapid spread of coronavirus disease 2019 (COVID-19) has raised concerns around the world. Since the first case was detected in Wuhan City, China, the disease has spread rapidly. The pathogen identified as a cause of COVID-19 is currently called severe acute respiratory syndrome coronavirus-2 (SARS-CoV-2) [1]; it has a phylogenetic resemblance to SARS-CoV-1 [2]. With a dramatic increase in daily confirmed global cases of COVID-19, the World Health Organization (WHO) declared a global pandemic on 12 March 2020 [3]. SARS-CoV-2 spreads by human-to-human transmission through droplets, the faecal-oral route, and direct contact and has an incubation period of 2–14 days [4].

Healthcare workers (HCWs) are at a higher risk of having COVID-19. According to the experience of the 2003 SARS outbreak, one-fifth of the global burden of SARS cases were healthcare workers. Several risk factors were identified during that time, including a lack of knowledge and preparedness, as well as poor infection control measures, lack of training, and poor compliance with the use of PPE while in contact with patients (suspected or not) and during high-risk procedures [5].

In the current pandemic, as of 21 April 2020, countries had reported to the World Health Organization (WHO) that over 35,000 HCWs were infected with COVID-19 [6]. In support of these established facts, further investigations on the aerodynamic nature of the virus revealed differences in the concentrations of SARS-CoV-2 RNA aerosols in different areas of two hospitals in Wuhan [2]. The areas with high load were those prone to crowds with carriers of the virus. Thus, healthcare workers are expected to be at a high risk of infection. Hazards include

pathogen exposure, long working hours, psychological distress, fatigue, occupational burnout, stigma, and physical and psychological violence.

One can recognize that the transmission of COVID-19 among HCWs is associated with overcrowding, the absence of isolation room facilities, and environmental contamination. However, this is likely compounded by the fact that some HCWs have inadequate knowledge of infection prevention practices [7]. Based on previous studies, the knowledge and attitudes of medical staff towards infectious diseases, and their willingness to work during an epidemic have been explored, including the knowledge and attitudes of critical care clinicians during the 2009 H1N1 influenza pandemic [8–10].

Protection of HCWs and prevention of intra-hospital transmission of infection are important aspects in epidemic responses, and this requires that HCWs must have updated knowledge regarding the source, transmission, symptoms, and preventive measures related to COVID-19. A previous study demonstrated that the level of knowledge on a particular of disease can influence attitudes and practices, and incorrect attitudes and practices directly increase the risk of infection. In general, lack of knowledge and misunderstandings among HCWs lead to delays in diagnosis, enhanced spread of the disease, and poor infection control practices [11].

Despite the extensive efforts made so far, accumulated evidence indicates that a poor understanding of the disease among HCW could result in delayed identification and treatment, leading to the rapid spread of infections. Moreover, health professionals are sacrificing their lives as a result of the pandemic, which incurs a significant cost to the global community [12]. Studies in different settings have revealed that there are huge differences in terms of the knowledge, awareness, attitude, practice, and preparedness of HCW in the fight against the pandemic [13–16].

While on duty in the recent COVID-19 pandemic, it has been reported that more than 10% of HCW have been infected with SARS-CoV-2. This demands more stringent measures to combat the pandemic and reduce mortality in this population. Consequently, the WHO has outlined the need for training HCW in order to reduce the rates of infection. However, research focusing on the assessment of knowledge, attitudes, and practices (KAP) of health professionals in this pandemic is very limited [13, 17].

In many ways, understanding HCWs KAP and possible risk factors can help to predict the outcomes of planned behaviours. If HCWs KAP concerning the virus and the factors that affect their attitudes and behaviours can be determined promptly in the early stages of an epidemic, then this information can provide relevant training and policies during the outbreak and guide HCWs in prioritizing protection and avoiding occupational exposure [18].

In the Ethiopian context, there are very few studies focusing on the KAP of health professionals regarding COVID-19. A previous study performed among nurses in northern Ethiopia demonstrated that the majority of health professionals had good knowledge, good infection prevention practices, and favourable attitudes [19]. In support of such evidence, a multi-centre study documented that a large majority of respondents had good knowledge and a positive attitude [14]. Despite their scarcity, previous studies were comparable and or slightly higher than a study conducted in an African setting, including Uganda and Nigeria, among this study population [15, 16].

Despite the high burden of the pandemic, there is scarce information regarding the knowledge and preparedness levels of HCW in Ethiopia. Therefore, the aim of this study was to assess knowledge and preparedness among HCW in Addis Ababa, Ethiopia. In the same context, we performed a study in the general population considering a KAP survey as a suitable format to evaluate existing programmes and to identify effective strategies for behaviour change in society and helps to predict outcomes of planned behaviour.

## Materials and methods

### Study design and study population

We used a cross-sectional survey to assess the knowledge and preparedness of health professionals in selected health facilities from March to April 2020 in Addis Ababa, Ethiopia. A total of six government hospitals and four private hospitals were included in the study. The participants were composed of medical doctors, nurses/midwives, pharmacists, and medical laboratory technologists/technicians. The inclusion criteria were being a health professional, adult (age > 18 years), willingness, and being on active duty during the data collection period. The Institutional Review Board (IRB) of the College of Health Sciences of Addis Ababa University approved the study protocol (Protocol number: 012/20/DMIP). A written informed consent was obtained from all participants. Participation was on a voluntary basis. This study was reported following the Strengthening the Reporting of Observational Studies in Epidemiology (STROBE) reporting guidelines.

### Sample size calculation and sampling method

A single population proportion formula was used with the assumption of 50% of health professionals had knowledge and preparedness for an epidemic, including COVID-19, and its management, with a 4% margin of error at a 95% confidence level, considering for a design effect of 2.0 and adding 15% for non-response, the study included 1, 334 health professionals.

### Sampling procedure

For this study, a multistage sampling was used. The first stage was stratifying health facilities by governmental and private ownership, and the second stage was cluster sampling among the two groups, taking a list of the facilities of each stratum. Six government hospitals and four private hospitals were included in the study through random selection. All health professionals were included in the study conveniently.

### Data collection

A standardized questionnaire adopted from a published protocol [20] was used. The questionnaire was initially developed in English (S1 File) for the intended purpose and a series of thorough revisions was carried out by a panel of experienced researchers in the field. The English version of the tool was translated to the local Amharic language (S2 File) and again back-translated to the English language to assure its consistency. A pre-test was done in 5% of the study participants to estimate the duration required to complete the survey, ensure clarity of the questions, avoid potential bias, and validate the data collection instrument. A Cronbach's alpha of 0.91 was used for preparedness and of 0.61 for awareness; however, the overall Cronbach's alpha of 0.65 indicated the acceptable validity (70–95%) of the questions [21].

The questionnaire had two sections. The first section was for general information, asking about age, sex, profession, department, the hospital where they work, and years of work experience as a health professional. The second section focused on the COVID-19 related knowledge and preparedness of the participants and the preparedness and working practices of respective hospitals.

The knowledge and preparedness assessments focus on personal and institutional issues. Personal issues include knowledge about signs and symptoms, identification of persons at risk, prevention measures, and tests recommended to confirm exposure to SARS-CoV-2. Additionally, the questions addressed issues regarding how the HCW were prepared on a personal level, if they knew how to use PPE, what to do if exposed, what to do if they developed signs

and symptoms and if they had knowledge on case management. At the institutional level, the questionnaire addressed if there was any triage protocol, isolation room, required equipment for case management if needed, risky medical procedures that generate aerosols, and if the chain of command was in place.

### Knowledge scoring system

By considering the total marks for each category, the score was graded as poor, moderate, or good based on cut-offs based on the modified Bloom cut-off point as follows: ≤12, poor;13–19, moderate; and ≥20, good [15].

### Data processing and analysis

First, the completeness of the data was checked, and data cleaning was carried out accordingly. Data entry was performed using EpiData Version 3.1and exported to SPSS software version 25 for analysis. Data analysis was done taking into consideration the type of the variables. Mean, standard deviation, and median were used (as appropriate) to summarize numerical data, whereas categorical data were summarized as frequencies and proportions. Inferential statistics for categorical variables were performed using the chi-square test of independence with Yates' continuity correction or Fisher's exact test, as appropriate.

## Results

### Socio-demographic characteristics

The study included 1334 health professionals whose ages ranged from 18 to 59 years, with a mean age of 30.71 ± 6.19 years. Around half (675, 50.7%) of the participants were female. The majority (532, 39.9%) of the participants were nurses/midwives, followed by doctors (397, 29.8%) and pharmacists (193, 14.5%). Table 1 shows the demographic characteristics of the study participants.

### Knowledge about COVID-19

Table 2 shows the details of the responses given by the health professionals for each knowledge question dealing with COVID-19 signs and symptoms, diagnostic modalities, related potential admission criteria required to identify patients at risk, and approaches to prevent the transmission in hospitals. The finding showed that over 80% of health professionals identified the correct response regarding knowledge about diagnostic techniques. With respect to diagnostic tests, 974(73%) subjects identified RT-PCR using respiratory samples as a diagnostic test for SARS-CoV-2 infection, whereas one- third (401, 30.1%) of subjects picked a serological test as a diagnostic test during the COVID-19 pandemic.

Regarding the identification of patients at risk of having COVID-19 upon hospital admission, 1181(88.5%) and 1014(76)% subjects identified travel to a COVID-19 affected area and contact with an infected person, respectively. Frequent hand washing, with soap and water, alcohol-based hand rubs and putting a mask on the face of known or suspected patients were identified by1251(93.4%) and 1214(91%) of health professionals, respectively. The mean knowledge score was 16.45±4.4 (range 2–25). Of the total,783 (58.7%), 354(26.5%), and 196 (14.7%) subjects had moderate, good, and poor knowledge of COVID-19, respectively.

With respect to prevention of transmission from known or suspected patients, health professionals knew most of the preventive measures. The majority of health professionals (1251, 93.8%) responded that hand washing with soap and water and hand rubbing with alcohol could be the possible ways to prevent COVID-19. The majority (1011, 75.8%) of them assumed

**Table 1. Demographic characteristics of health professionals.**

| Demographic characters | | Number | Percent |
|---|---|---|---|
| **Sex** | Male | 656 | 49.3 |
| | Female | 675 | 50.7 |
| **Age group (years)** | Mean age, 30.71 ± 6.19 years | | |
| | ≤24 | 116 | 8.7 |
| | 25–29 | 546 | 40.9 |
| | 30–34 | 374 | 28.0 |
| | 35–39 | 142 | 10.6 |
| | ≥40 | 108 | 8.1 |
| **Profession** | Doctor | 397 | 29.8 |
| | Pharmacist | 193 | 14.5 |
| | Nurse/midwives | 532 | 39.9 |
| | Medical laboratory | 207 | 15.5 |
| **Hospitals** | Government hospitals | 802 | 60.1 |
| | Private hospitals | 532 | 39.9 |
| **Service years of employment** | ≤5 | 791 | 59.3 |
| | >5 | 456 | 34.2 |

that eating cooked and boiled food was protective in the fight against COVID-19. The majority (1214, 91%) of subjects responded that putting a mask on suspected or known patients prevents the transmission of SARS-CoV-2.

This study explored knowledge levels across professions, as well as in those with and without formal training. Accordingly, those with formal training had moderate knowledge, which represented doctors (95, 69.3%), nurses/midwives (106, 69.3%), pharmacists (34, 64.2%), and medical laboratory staff (36, 65.5%). However, those without training had slightly lower moderate knowledge levels. As shown in Table 3, the study assessed the association of knowledge of health professionals with demographic characteristics.

## Preparedness of health professionals regarding the pandemic

A detailed assessment of the preparedness of health professionals and the hospital they work at is shown in Table 4. The assessment of the preparedness of health professionals provided mixed results, with fewer than 50% of health professionals using precautionary measures during risky procedures, without guidelines where to report a potential case or exposure, or criteria that guide the evaluation of persons under investigation.

Among the total, 220(56.8%) doctors, 127(67.2%) pharmacists, 323(61.5%) nurses/midwives, and 119(58.6%) medical laboratory professionals responded that they were prepared for COVID-19 management (p-value < 0.10). Out of the total,287(73.6%) doctors, 110(57.3%) pharmacists, 303(57.5%) nurses/midwives, and 131(63.9%) medical laboratory professionals were confident enough on how to use PPE in case of possible contact with COVID-19 patients (p-value < 0.001). The finding revealed that the majority of HCWs were not confident in handling suspected COVID-19 patients (p-value < 0.07).

## Assessment of hospital preparedness by health professionals

Regarding the preparedness of the hospitals, approximately 50% of health professionals thought that their respective hospital was prepared for COVID-19 (p-value <0.001). The study demonstrated that close to 50% were unsatisfied with the medical equipment available for

**Table 2. The proportions of correct answers about the signs, diagnostic methods, identification criteria, and prevention measures regarding COVID-19 given by health professionals.**

|  | Number | Percent |
|---|---|---|
| **K1: Signs and symptoms of COVID-19** | | |
| Fever | 1304 | 97.8 |
| Cough | 1278 | 95.8 |
| Sneezing | 977 | 73.2 |
| Runny nose | 658 | 49.3 |
| Sore throat | 1124 | 84.3 |
| Shortness of breath | 1096 | 82.2 |
| Pressure/pain in the chest | 486 | 36.4 |
| Joint/muscle pain | 583 | 43.7 |
| Red eyes | 237 | 17.8 |
| Rash | 166 | 12.4 |
| Diarrhoea | 493 | 37.0 |
| May present without symptoms | 394 | 29.5 |
| **K2: COVID-19 diagnostics** | | |
| RT-PCR with respiratory samples | 974 | 73.0 |
| RT-PCR with serum samples | 768 | 57.6 |
| Chest X-ray | 501 | 37.6 |
| Serological tests | 401 | 30.1 |
| **K3: Identification criteria for patients at risk for COVID-19** | | |
| Presence of diarrhoea | 435 | 32.6 |
| Respiratory infection symptoms | 986 | 73.9 |
| Travel to COVID-affected areas | 1181 | 88.5 |
| Contact with possible infected patients | 1014 | 76.0 |
| **K4: Measures for prevention of transmission from known or suspected patients** | | |
| Frequent hand washing with soap and water/alcohol-based hand rub | 1251 | 93.8 |
| Avoiding eating uncooked food | 1011 | 75.8 |
| Wearing a face mask | 1214 | 91.0 |
| Placing known or suspected patients in adequately ventilated single rooms | 1161 | 87.0 |
| Wearing protective clothing | 1144 | 85.8 |
| Avoiding moving and transporting patients | 1105 | 82.8 |
| Routine cleaning and disinfecting surfaces | 1075 | 80.6 |

COVID-19 treatment in their hospitals (p-value < 0.002). However, 57.4% indicated that their hospital had an established triage protocol.

## COVID-19 related sources of information

Media (television, radio, and newspapers) were the primary sources of information for most of the health professionals (1020, 76.5%), followed by social networks (Facebook, Twitter, and blogs) accounting for 899(67.4%) subjects. Among the total, 398 (29.8%) subjects had received formal training regarding the COVID-19 outbreak and 843 (63.2%) subjects responded that their hospital took measures to keep staff updated on COVID-19.

## Discussion

Success in the fight against COVID-19 is dependent on the involvement of the public, HCW, and the appropriate actions by the government. Although the general public has been

**Table 3. Association of knowledge with demographic characteristics.**

| Characteristics | | Knowledge/knowledge scores | | | p-value |
|---|---|---|---|---|---|
| | | Poor | Moderate | Good | |
| | | N (%) | N (%) | N (%) | |
| **Sex** | Male | 79 (12.1) | 366 (55.9) | 210 (32.1) | < 0.001 |
| | Female | 117 (17.3) | 415 (61.5) | 143 (21.2) | |
| **Age group (years)** | ≤24 | 21 (18.1) | 75 (64.7) | 20 (17.2) | 0.003 |
| | 25–29 | 89 (16.3) | 312 (57.2) | 144 (26.4) | |
| | 30–34 | 44 (11.8) | 234 (62.6) | 96 (25.7) | |
| | 35–39 | 22 (15.5) | 70 (49.3) | 50 (35.2) | |
| | ≥40 | 6 (5.6) | 72 (66.7) | 30 (27.8) | |
| **Profession** | Doctor | 18 (4.5) | 214 (54.0) | 164 (41.4) | < 0.001 |
| | Pharmacist | 48 (24.9) | 111 (57.5) | 34 (17.6) | |
| | Nurse/midwife | 98 (18.4) | 333 (62.6) | 101 (19.0) | |
| | Medical laboratory | 31 (15.0) | 123 (59.4) | 53 (25.6) | |

**Table 4. Assessment of preparedness of health professionals and the respective hospitals towards the pandemic.**

| Statements | Yes | No |
|---|---|---|
| | N (%) | N (%) |
| Self-preparedness indicator | | |
| Do you think that you have the latest information on COVID-19? | 969 (72.6) | 340 (25.5) |
| Do you consider yourself prepared for the management of the 2019-nCoV outbreak? | 793 (59.4) | 516 (38.7) |
| In case of contact with possible 2019-nCoV patients, do you know how to use personal protective equipment (PPE)? | 834 (62.5) | 485 (36.4) |
| In case of contact with confirmed 2019-nCoV patients, do you know how to perform isolation procedures on the patients? | 735 (55.1) | 581 (43.6) |
| Do you know the precautionary measures to take when performing aerosol-generating procedures? | 567 (42.5) | 741 (55.5) |
| Do you know the criteria to guide the evaluation of persons under investigation (PUI) for COVID-19? | 573 (43.0) | 750 (56.2) |
| Do you know where to take the report form, and how to report a potential 2019-nCoV case or exposure to facility infection control leaders and public health officials? | 723 (54.2) | 599 (44.9) |
| Do you know who to contact (chain of command) in outbreak situations in your hospital? | 742 (55.6) | 573 (43.0) |
| Do you know what to do if you have signs or symptoms of suspected 2019-nCoV infection? | 1079 (80.9) | 238 (17.8) |
| Do you know who to contact in a situation where there has been an unprotected exposure to a known or suspected 2019-nCoV patient? | 666 (49.9) | 646 (48.4) |
| Hospital preparedness measures | | |
| Is there an established sort of protocol for triage and isolation in your hospital regarding COVID-19? | 766 (57.4) | 183 (13.7) |
| Has your hospital made available an airborne infection isolation room (AIIR)? | 613 (46.0) | 225 (16.9) |
| Do you consider your hospital prepared for the management of the 2019-nCoV outbreak? | 641 (48.1) | 319 (23.9) |
| Has your hospital established procedures for controlling visitors to known or suspected 2019-nCoV patients? | 580 (43.5) | 187 (14.0) |

informed to stay at home, HCWs go to their clinics and hospitals. Experience from China and Italy indicates thatthat there were reported deaths of over 3,300 (20%) HCWs [22]. Those who were not infected or survived suffered from physical and mental exhaustion. Alongside their personal safety concerns, HCWs were worried about their families, as well as their patients.

Despite its public health burden and economic impact, there is an ample knowledge deficit among health professionals. Unless efforts are made to improve the knowledge of health professionals and maximize their preparedness, the pandemic will have a tremendous impact on the healthcare system and consequently could alter COVID-19 prevention and management. At this time of crisis, assessing the knowledge and preparedness of health professionals is helpful to identify gaps and correct them in a timely manner.

According to a previous study, young age and few service years have been considered as important factors that could place an individual at a higher risk of infection. Our findings indicated that most participants were in the age range of 25–29 years and had served for < 5 years, which agrees with a study from Nigeria. Such characteristics were a major source of infection and a means of spreading infection [23].

Globally, a wide range of difference was noted in knowledge and preparedness regarding COVID-19 among health professionals. However, knowledge can in many ways impact attitudes, behaviours, and an individual's positive attitude, and consequently, change behaviour in a broader context [11]. Our findings demonstrated that two-thirds of the participants (58.7%) had moderate knowledge.

The overall knowledge of the participating HCWs regarding signs and symptoms, identification of persons at risk of developing disease, case definition of COVID-19, appropriate tests offered to suspected cases and high-risk patients, and preventive measures that help to minimize the risk of transmission of SARS-CoV-2 was good. Unfortunately, 40–60% of the HCW neither knew whom to contact in a hospital outbreak situation or upon unprotected exposure or did they know the criteria to guide the evaluation of suspected cases and how to perform isolation procedures. Additionally, 55% of the HCW did not know about precautionary measures when performing aerosol-generating procedures, despite claiming they had the latest information.

A comparison was made with previously conducted research among a similar study population. A study conducted in the early phase of the pandemic in Ethiopia showed that HCW perceived that they were not yet well-prepared and felt unable to respond to the pandemic quickly and efficiently [24]. The current finding was comparable with a study conducted in Uganda, where they had sufficient knowledge (69%), a positive attitude, and good practices regarding COVID-19 [15]. Our finding was slightly higher than those of a study done in Nigeria, where the majority (168, 56%) of the participants were highly aware of the pandemic [16].

In line with our study, a study performed in Iran indicated that more than half of health professionals (56.5%) had good knowledge about the sources, transmission, symptoms, signs, prognosis, treatment, and mortality rate of COVID-19 [25]. Contrary to our findings, another study from Iran revealed that HCWs had insufficient knowledge about COVID-19 but had positive perceptions of the prevention of COVID-19 transmission [26].

Some of our study findings were far lower than studies performed elsewhere. In a previous study in Ethiopia, most of the participants (74%) had good knowledge [19]. Our results were also lower thana study from Egypt, where the mean correct answer rate was 80.4% [27]. Another study from China demonstrated that a high proportion (89%) of HCWs had sufficient knowledge of COVID-19. In their study, similar to our findings, they indicated that doctors showed higher knowledge scores (38.56 ± 3.31) compared to those in nurses (37.85 ± 2.63) and paramedics (36.72 ± 4.82) [28]. Moreover, our study result was lower than those of a study performed in China, which showed 88.4% of participants possessed sufficient knowledge [29].

A study from Pakistan documented that HCW had good knowledge (93.2%), positive attitudes, and good practices (88.7%) [30]. In support of the findings, a study from Greece documented that most of the respondents (88.28%) had a good level of knowledge [13]. Although our findings in Ethiopia are encouraging, still enhancing the knowledge status of health professionals seems possible when comparing to these other similar settings.

This study explored the overall mean knowledge score, which was 16.45±4.4 (range 2–25). Our finding was in line with the overall knowledge score reported from Iran [15] and a mean knowledge score of 18.5 ± 2.7 in Egypt [27]. These results were higher by far compared to a report from Vietnam that reported a mean score of knowledge of 8.17±1.3. Their findings showed that HCWs had a high level of knowledge and a positive attitude towards the COVID-19 outbreak [29]. However, much work must be done to better mitigate the challenges and promote the safety of health professionals.

In our findings, the specific knowledge of the participants on different aspects of COVID-19 was investigated. Interestingly, the majority identified the signs and symptoms, diagnostic modalities, prevention measures, and admission criteria for patients who are at risk for COVID-19. Accordingly, most of the participants were able to identify fever, cough, sneezing, and sore throat as the major signs and symptoms of COVID-19.

In agreement with our findings, a study from Iran revealed that more than half of the nurses (56.5%) had good knowledge about the sources, transmission, symptoms, signs, prognosis, treatment, and mortality rate of COVID-19 [25]. Unlike our finding, a study from the United Arab Emirates revealed that a significant proportion of HCW had poor knowledge of transmission (61%) and symptoms onset (63.6%) [26]. However, less than the majority incorrectly identified signs and symptoms not related to the pandemic. Such evidence is critical for outlining major gaps in this topic for the delivery of the latest information to this group and to enhance the accurate clinical diagnosis of the disease. In many ways, this could have an impact on early detection, the rate of transmission, and prevention of infection.

With respect to understanding the possible diagnostic modalities, our study showed that the majority were aware of the samples to be collected and the respective diagnostic platform for COVID-19 diagnosis. This is interesting and encouraging because this will have an impact on early detection, management of patients, infection prevention, and control of the pandemic. Considering the dynamics of the pandemic, aggressive efforts must be made to provide up-to-date information on the type of samples, procedures for appropriate sample collection, principles of the methods, limitations, and interpretations of the findings. Although Ethiopia was not using serological testing, the poor knowledge of participants in this respect should urge stakeholders to provide comprehensive information about alternative laboratory diagnostic approaches for COVID-19 in the local context.

Currently, to fight against COVID-19, the firm application of prevention protocols are necessary at all levels. With respect to knowledge on preventive measures from known or suspected patients, 93.8% of the health professionals mentioned hand washing with soap and water and hand rubbing with alcohol as an important prevention measures against COVID-19. In line with this finding, a study from northern Ethiopia indicated that the participants had good infection prevention practices with a favourable attitude [19]. Since the primary means of containing the pandemic is through prevention, having such an understanding is critical because this measure could possibly break the transmission chain of SARS-CoV-2. Health professionals should be provided with the latest information on infection prevention and control.

Additionally, one of the most important aspects of an outbreak is the identification of patients who have symptoms and are at a high risk of having the disease. With this regard, the assessment showed that most healthcare professionals recognized the identification criteria of patients with COVID-19. Basically, such understanding enables health professionals to identify

cases at an early stage to establish appropriate management and minimize the spread of SARS-CoV-2 infection. With continuous support from concerned stakeholders, health professionals can combat the current pandemic and any possible outbreak in the future. Therefore, health professionals should be provided with the latest training in all aspects of COVID-19, including prevention modalities, transmission mode, diagnostic strategies, prevention strategies, management of cases, what to do following exposure, and the chain of command for reporting unusual events to contain the pandemic.

In general, having sufficient knowledge may reflect the successful dissemination of information about COVID-19 by different media. In this regard, this study explored from where were health professionals obtaining health-related information on COVID-19. Accordingly, media (television, radio, and newspapers) were the primary sources of information, followed by social networks (Facebook, Twitter, and blogs). This could be explained by the high rate of transmission of COVID-19 around the world, which might have increased the attention of health professionals and their subsequent knowledge of this pandemic. In the same context, in Africa, a recent study demonstrated that the most common source of information was through colleagues (143, 47.67%) [16].

In agreement with our findings, a study from Iran indicated a stunning figure—60% of HCW used social media as a source of information [26]. In this respect, another finding indicated that the sources of information for nurses were the WHO and the Ministry of Health (55.3%), social networks (48.23%), and media (42.35%) [25], which were somehow credible and reliable sources of information unlike the others. Comparably, a study from Vietnam demonstrated that the main sources of COVID-19 information were social media and the Ministry of Health website (91.1% and 82.6%, respectively) [29].

Another study conducted in a low-resource setting showed that most participants (70%) used social media as a source of information on COVID-19 [31]. In our study, we did not identify national guidelines or WHO websites as a source of information, which is major gap that requires immediate attention. The widespread use of the internet and its availability to wider sectors of society has made it a major source of information. Although information from social media has had a positive impact on the prevention and control of the disease, there should be regulation to minimize/avoid misinformation and combat the current situation in the right manner.

This baseline study is limited by its cross-sectional design. Additionally, the 10 hospitals were selected randomly considering the patient flow. Although the total sample size of the current study is more or less satisfactory, we believe that it would have been better if much more participants are recruited to the study. Additionally, the study was conducted only in Addis Ababa, where people had plenty of access to life-saving health related information on various diseases and the current pandemic as well. Consequently, the findings only reflected a sneak peak of the situation in Ethiopia. However, the study will serve as a guide for planning and implementing interventions targeted at controlling epidemics.

## Conclusion

This study concluded that health professionals had moderate knowledge regarding the COVID-19 pandemic. However, still showing a need for more information despite high input from the Ministry of Health and WHO in Ethiopia. Younger staff, non-physicians, and females need to specifically be addressed and included in trainings. The preparedness of health professionals towards COVID-19 was encouraging in many aspects, concerning knowledge of symptoms, diagnostic methods, and handling of patients. Around half of the staff felt their institution was prepared and had guidelines, allocated specific areas, and provided equipment

and consumables. Given the lack of infection control items globally, hospital preparedness is expected to a lesser extent in the low-resource setting. This calls towards global solidarity and innovative approaches to assure the safety of staff and patients.

## Supporting information

**S1 File. Health professional preparedness: Questionnaire.**
(PDF)

**S2 File.**
(DOCX)

## Acknowledgments

We would like to thank all study participants and data collectors. Our appreciation goes to Addis Ababa University and the College of Health Sciences management for all their support.

## Author Contributions

**Conceptualization:** Zelalem Desalegn, Negussie Deyessa, Damen Hailemariam, Adamu Addissie, Abdulnasir Abagero, Tamrat Abebe.

**Data curation:** Zelalem Desalegn, Negussie Deyessa, Brhanu Teka, Welelta Shiferaw, Meron Yohannes, Adamu Addissie, Abdulnasir Abagero, Mirgissa Kaba, Workeabeba Abebe, Alem Abrha, Wondimu Ayele, Tewodros Haile, Yirgu Gebrehiwot, Wondwossen Amogne, Eva Johanna Kantelhardt, Tamrat Abebe.

**Formal analysis:** Zelalem Desalegn, Brhanu Teka, Welelta Shiferaw, Adamu Addissie, Abdulnasir Abagero, Workeabeba Abebe, Berhanu Nega, Wondimu Ayele, Tewodros Haile, Yirgu Gebrehiwot, Wondwossen Amogne, Eva Johanna Kantelhardt, Tamrat Abebe.

**Funding acquisition:** Zelalem Desalegn, Negussie Deyessa, Damen Hailemariam, Adamu Addissie, Tamrat Abebe.

**Investigation:** Zelalem Desalegn, Negussie Deyessa, Brhanu Teka, Welelta Shiferaw, Meron Yohannes, Damen Hailemariam, Adamu Addissie, Abdulnasir Abagero, Mirgissa Kaba, Workeabeba Abebe, Alem Abrha, Wondimu Ayele, Tewodros Haile, Yirgu Gebrehiwot, Wondwossen Amogne, Eva Johanna Kantelhardt, Tamrat Abebe.

**Methodology:** Zelalem Desalegn, Negussie Deyessa, Brhanu Teka, Welelta Shiferaw, Damen Hailemariam, Adamu Addissie, Abdulnasir Abagero, Mirgissa Kaba, Eva Johanna Kantelhardt, Tamrat Abebe.

**Project administration:** Zelalem Desalegn, Brhanu Teka, Welelta Shiferaw, Adamu Addissie, Tamrat Abebe.

**Resources:** Zelalem Desalegn, Tamrat Abebe.

**Software:** Zelalem Desalegn, Tamrat Abebe.

**Supervision:** Zelalem Desalegn, Negussie Deyessa, Brhanu Teka, Meron Yohannes, Damen Hailemariam, Adamu Addissie, Abdulnasir Abagero, Workeabeba Abebe, Alem Abrha, Berhanu Nega, Tewodros Haile, Tamrat Abebe.

**Validation:** Zelalem Desalegn, Negussie Deyessa, Brhanu Teka, Welelta Shiferaw, Meron Yohannes, Damen Hailemariam, Adamu Addissie, Abdulnasir Abagero, Mirgissa Kaba,

Workeabeba Abebe, Alem Abrha, Berhanu Nega, Wondimu Ayele, Tewodros Haile, Yirgu Gebrehiwot, Wondwossen Amogne, Eva Johanna Kantelhardt, Tamrat Abebe.

**Visualization:** Zelalem Desalegn, Negussie Deyessa, Brhanu Teka, Welelta Shiferaw, Meron Yohannes, Damen Hailemariam, Adamu Addissie, Abdulnasir Abagero, Mirgissa Kaba, Workeabeba Abebe, Alem Abrha, Berhanu Nega, Wondimu Ayele, Tewodros Haile, Yirgu Gebrehiwot, Wondwossen Amogne, Eva Johanna Kantelhardt, Tamrat Abebe.

**Writing – original draft:** Zelalem Desalegn, Brhanu Teka, Welelta Shiferaw, Wondimu Ayele, Tamrat Abebe.

**Writing – review & editing:** Zelalem Desalegn, Negussie Deyessa, Brhanu Teka, Welelta Shiferaw, Meron Yohannes, Damen Hailemariam, Adamu Addissie, Abdulnasir Abagero, Mirgissa Kaba, Workeabeba Abebe, Alem Abrha, Berhanu Nega, Wondimu Ayele, Tewodros Haile, Yirgu Gebrehiwot, Wondwossen Amogne, Eva Johanna Kantelhardt, Tamrat Abebe.

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
