## [Decision Letter · Decision Letter 0]

22 Jul 2020

PONE-D-20-19732

Evaluation of COVID-19 Related Health Professionals Knowledge and Preparedness in Selected Health Facilities in resource limited setting Addis Ababa, Ethiopia

PLOS ONE

Dear Dr. Desalegn,

Thank you for submitting your manuscript to PLOS ONE. After careful consideration, we feel that it has merit but does not fully meet PLOS ONE’s publication criteria as it currently stands. Therefore, we invite you to submit a revised version of the manuscript that addresses the points raised during the review process.

We look forward to receiving your revised manuscript.

Kind regards,

Khin Thet Wai, MBBS, MPH, MA (Population & Family Planning Resear

Academic Editor

PLOS ONE

Additional Editor Comments:

1. Authors need to disclose the information on the first manuscript submitted to the journal which is related to COVID-19 infection in the same local context.

2. Grammatical errors throughout the manuscript and lack of clarity in most of the paragraphs require extensive language editing.

3. The main claims of the paper are focused on knowledge of healthcare workers and their reported preparedness of health care facilities to manage COVID-19 infection. Authors need to add more description on how significant are those claims for the improvement of service delivery and program operations at health facilities in the Introduction by citing appropriate literature.

4. Importantly, authors need to follow Strobe Guidelines for Observational Studies in the writeup of methods section and discussion. Limitations of the study are missing.

5. It is critical to note that the conclusion section is weak and requires a revision.

Journal Requirements:

4. Please include a copy of Table 4 which you refer to in your text on page 19.

Reviewers' comments:

Reviewer's Responses to Questions

**Comments to the Author**

1. Is the manuscript technically sound, and do the data support the conclusions?

Reviewer #1: Yes

Reviewer #2: Yes

2. Has the statistical analysis been performed appropriately and rigorously? 

Reviewer #1: Yes

Reviewer #2: Yes

3. Have the authors made all data underlying the findings in their manuscript fully available?

Reviewer #1: No

Reviewer #2: Yes

4. Is the manuscript presented in an intelligible fashion and written in standard English?

Reviewer #1: Yes

Reviewer #2: Yes

5. Review Comments to the Author

Reviewer #1: A timely manuscript evaluating the knowledge and preparedness of HCWs in Addis.

some minor comments

1- Recent work on KAPs of HCWs in Africa and beyond has been published. Authors should search for these papers and cite them as appropriate.

2. Method. Remove the number of participants from the study design (keep it only in the results section)

3. Use recent data to describe the global and local burden of COVID-19

Reviewer #2: The corrections are done in the manuscript. The author should check the manuscript and effect all the noted corrections.

The author should effects the Minor revisions and return the manuscript for final evaluation and publication.

It is important the author consult the work done by Ogolodom et al as recommended. It will help them seriously.

6. PLOS authors have the option to publish the peer review history of their article (what does this mean?). If published, this will include your full peer review and any attached files.

Reviewer #1: **Yes: **Felix Bongomin, MD,MSc

Reviewer #2: **Yes: **MICHAEL PROMISE OGOLODOM

---

## [Author Response · Author response to Decision Letter 0]

25 Sep 2020

Response to Editor and reviewers comments

1. Editor comments

1. Authors need to disclose the information on the first manuscript submitted to the

journal which is related to COVID-19 infection in the same local context.

Well taken. Based on the comment, we have considered research based information from the local context considered in the manuscript. This has happened in the introduction part and while discussing our findings. We appreciate your comment.

2. Grammatical errors throughout the manuscript and lack of clarity in most of the

paragraphs require extensive language editing.

Well taken. The grammatical errors and lack of clarity addressed in the revised version of the manuscript. Additionally, the revised version has been sent out for language proof-reading service and corrections were accommodated in the revised version of the manuscript. However, if there are any unaddressed, the authors are highly interested and ready to correct them as per the requirements and the scientific standard. 

3. The main claims of the paper are focused on knowledge of healthcare workers and their reported preparedness of health care facilities to manage COVID-19 infection. Authors need to add more description on how significant are those claims for the improvement of service delivery and program operations at health facilities in the Introduction by citing appropriate literature.

Well taken. Accordingly, we have provided data demonstrating the relevance of doing research related with the current topic in the improvement of service delivery and program operations at health facilities. 

4. Importantly, authors need to follow Strobe Guidelines for Observational Studies in the

write up of methods section and discussion. Limitations of the study are missing.

Well taken. STROBE guidelines for any cross-sectional study considered in the method section

5. It is critical to note that the conclusion section is weak and requires a revision.

Well taken. The conclusion section re-considered and written based the finding of the research finding. The change has been shown in the revised manuscript with track changes.

6. Please include a copy of Table 4 which you refer to in your text on page 19.

We appreciate your comments. We have included the copy of Table 4 in the text.

Journal Requirements:

When submitting your revision, we need you to address these additionalrequirements.

7. Please ensure that your manuscript meets PLOS ONE's style requirements, including

those for file naming.

Well taken. The authors believe that our manuscript meets the PLOS ONE's style requirements.

However, the authors are eager to address them with your directions if there are any unnoticed points in the revised version.

8. Please include additional information regarding the survey or questionnaire used in the study and ensure that you have provided sufficient details that others could replicatethe analyses. For instance, if you developed a questionnaire as part of this study and it is not under a copyright more restrictive than CC-BY, please include a copy, in both the original language and English, as Supporting Information.

Well taken. We have included the data collection instrument.

9. PLOS requires an ORCID iD for the corresponding author in Editorial Manager on

papers submitted after December 6th, 2016. Please ensure that you have an ORCID iD

and that it is validated in Editorial Manager.

Well taken. The submitting author has created the iD and validated in the Editorial manager 

2. Reviewer Comments to the Author

Dear Reviewers, 

The authors appreciate the very positive feed backs of the reviewers forwarded to the very relevant research questions. Our heartfelt thanks goes to the reviewers for their additional inputs and specific comments raised at different section of the manuscript. 

Reviewer #1: A timely manuscript evaluating the knowledge and preparedness ofHCWs in Addis.

some minor comments

1- Recent work on KAPs of HCWs in Africa and beyond has been published. Authors

should search for these papers and cite them as appropriate.

Well taken. Research works which have been conducted in African context and beyond searched for the purpose to revealing the real picture and strengthening the manuscript further. A very recent research findings were considered in the introduction section for outlining what has been known so far in the area of the research problem. Additionally, we have used plenty of research findings while comparing our finding in the discussion section. 

2. Method. Remove the number of participants from the study design (keep it only in the

results section)

Well taken. The number of participants removed from the study design section.

3. Use recent data to describe the global and local burden of COVID-19

We appreciate the comment. Well taken. Since an ample of researches based evidences are coming out and figures are changing from time to time, we accept of considering very latest data to describe COVID 19 in the local and global context and accommodated accordingly.. 

Reviewer #2:The corrections are done in the manuscript. The author should check the

manuscript and effect all the noted corrections.

1-The author should effects the Minor revisions and return the manuscript for finalevaluation and publication.

Well taken. Every given comments checked in the manuscript and considered in the revised version of the manuscript

2-It is important the author consult the work done by Ogolodom et al as recommended. Itwill help them seriously.

Well taken. On top of considering additional research data released at the local and global level, we were able to use the recommended research work done by Ogolodom et al. As you have mentioned, we found the paper very helpful for our purpose and used in the introduction section and discussion part of the revised manuscript.

---

## [Decision Letter · Decision Letter 1]

7 Oct 2020

PONE-D-20-19732R1

Evaluation of COVID-19 related knowledge and preparedness in health professionals at selected health facilities in a resource-limited setting in Addis Ababa, Ethiopia

PLOS ONE

Dear Dr. Desalegn,

Thank you for submitting your manuscript to PLOS ONE. After careful consideration, we feel that it has merit but does not fully meet PLOS ONE’s publication criteria as it currently stands. Therefore, we invite you to submit a revised version of the manuscript that addresses the points raised during the review process.

Specifically, we notice some errors in English language usage and typos in your manuscript. As PLOS ONE does not provide copy editing or proofs of accepted manuscript, we therefore recommend that you carefully review your manuscript and correct any errors at this time.

We look forward to receiving your revised manuscript.

Kind regards,

Khin Thet Wai, MBBS, MPH, MA (Population & Family Planning Resear

Academic Editor

PLOS ONE

Journal Requirements:

Please carefully review your manuscript and correct any language errors at this time.

Reviewers' comments:

Reviewer's Responses to Questions

**Comments to the Author**

1. If the authors have adequately addressed your comments raised in a previous round of review and you feel that this manuscript is now acceptable for publication, you may indicate that here to bypass the “Comments to the Author” section, enter your conflict of interest statement in the “Confidential to Editor” section, and submit your "Accept" recommendation.

Reviewer #1: All comments have been addressed

Reviewer #2: All comments have been addressed

2. Is the manuscript technically sound, and do the data support the conclusions?

Reviewer #1: Yes

Reviewer #2: Yes

3. Has the statistical analysis been performed appropriately and rigorously? 

Reviewer #1: Yes

Reviewer #2: Yes

4. Have the authors made all data underlying the findings in their manuscript fully available?

Reviewer #1: No

Reviewer #2: Yes

5. Is the manuscript presented in an intelligible fashion and written in standard English?

Reviewer #1: Yes

Reviewer #2: Yes

6. Review Comments to the Author

Reviewer #1: The authors have address all the comments of the reviewers and the manuscript can now be considered for publication by the Academic Editor

Reviewer #2: (No Response)

7. PLOS authors have the option to publish the peer review history of their article (what does this mean?). If published, this will include your full peer review and any attached files.

Reviewer #1: **Yes: **Felix Bongomin, MD

Reviewer #2: **Yes: **Michael Promise Ogolodom

---

## [Author Response · Author response to Decision Letter 1]

3 Nov 2020

Comment: The manuscript requires correction of grammatical and spelling error. We have thoroughly gone through the manuscript to assure English spelling, grammar, numbering and spacing. Additionally, as per the forwarded comment, the revised version was send out for proof reading service. Accordingly, the necessary correction was made. 

Evidence: The proof reading service certificate was uploaded along with the other files.

---

## [Decision Letter · Decision Letter 2]

13 Nov 2020

PONE-D-20-19732R2

Evaluation of COVID-19 related knowledge and preparedness in health professionals at selected health facilities in a resource-limited setting in Addis Ababa, Ethiopia

PLOS ONE

Dear Dr. Desalegn,

Thank you for submitting your manuscript to PLOS ONE. After careful consideration, we feel that it has merit but does not fully meet PLOS ONE’s publication criteria as it currently stands. Therefore, we invite you to submit a revised version of the manuscript that addresses the points raised during the review process.Please submit your revised manuscript by Dec 28 2020 11:59PM. If you will need more time than this to complete your revisions, please reply to this message or contact the journal office at plosone@plos.org. Please include the following items when submitting your revised manuscript:

We look forward to receiving your revised manuscript.

Kind regards,

Khin Thet Wai, MBBS, MPH, MA (Population & Family Planning Resear

Academic Editor

PLOS ONE

Additional Editor Comments (if provided):

1. Authors have adequately addressed the comments of reviewers.

2. Authors need to follow the journal guidelines in preparing the Reference section.

3. Still, there is a need to correct some grammatical errors to improve readability and to meet the journal standard.

Abstract-

LINE 69: "The current study revealed that" (to replace shows)

Text

LINE 122: "Despite the extensive efforts made so far, accumulated evidence indicates that" (to remove shows)

LINE 160: "considering a KAP survey as a suitable format'

LINE 157: " A total of six government hospitals and four private hospitals were included in the study"

LINE 162: "A written informed consent"

LINE 168: "health professionals had knowledge and preparedness"

LINE 169: "considering for a design effect of"

LINE 218: "The study included 1,334 health professionals"

LINE 229: "The finding showed that"

LINE 253: "represented doctors"

LINE 302: please correct as follows- "that there were reported deaths of over 3,300 (20%) HCWs [22]."

LINE 304: "HCWs were worried"

LINES 308-309: Please correct as follows- "At this time of crisis" rather than "At a time like this".

LINE 319: "Our findings demonstrated that"

LINE 417: "which were somehow"

LINE 426: "there should be a regulation"

LINE 432: "participants are recruited to the study"

LINE 433: "where people had plenty of access"

LINE 434: "consequently, the findings only reflected a sneak peak"

LINE 438: "This study concluded that"

Reviewers' comments:

Reviewer's Responses to Questions

**Comments to the Author**

1. If the authors have adequately addressed your comments raised in a previous round of review and you feel that this manuscript is now acceptable for publication, you may indicate that here to bypass the “Comments to the Author” section, enter your conflict of interest statement in the “Confidential to Editor” section, and submit your "Accept" recommendation.

Reviewer #1: All comments have been addressed

Reviewer #2: (No Response)

2. Is the manuscript technically sound, and do the data support the conclusions?

Reviewer #1: Yes

Reviewer #2: (No Response)

3. Has the statistical analysis been performed appropriately and rigorously? 

Reviewer #1: Yes

Reviewer #2: (No Response)

4. Have the authors made all data underlying the findings in their manuscript fully available?

Reviewer #1: Yes

Reviewer #2: (No Response)

5. Is the manuscript presented in an intelligible fashion and written in standard English?

Reviewer #1: Yes

Reviewer #2: (No Response)

6. Review Comments to the Author

Reviewer #1: The authors have sufficiently addressed all comments by the reviewer. The manuscript is now acceptable in its current form. Well done !

Reviewer #2: I am okay with all the corrections done by the author.

7. PLOS authors have the option to publish the peer review history of their article (what does this mean?). If published, this will include your full peer review and any attached files.

Reviewer #1: **Yes: **Felix Bongomin, MD, MSc (Gulu University, Uganda)

Reviewer #2: No

---

## [Author Response · Author response to Decision Letter 2]

2 Dec 2020

Response to reviewers

Manuscript number: PONE-D-20-19732

Manuscript title: Evaluation of COVID-19 related knowledge and preparedness in health professionals at selected health facilities in a resource-limited setting in Addis Ababa, Ethiopia

Dear editor and the reviewer, 

We are very grateful for the valuable comments and scientific guidance forwarded from the editor, Plos One journal team member and the respective reviewers. We have learnt a lot in the process of the revision process which would have great impact in the personal and professional development. Dear academic editor and reviewers, your inputs were highly critical in enriching the scientific paper to utmost standard. 

Dear reviewers, we appreciate your commitment taken of reviewing the manuscript aimed exploring the response of health professionals toward COVID-19. Your comments were well taken and incorporated accordingly in the revised version of the manuscript. 

We have gone through the document and the respective additions, modification, elaboration and correction made whenever the need arise as per the comments concern. If there are anything to which the editor and the reviewers are not satisfied/ and or clear with, the authors are highly eager and open-minded to entertain the missed points at any time point. 

In the online submission, we uploaded documents including [1. Manuscript without track changes [2. Manuscript with track changes [3. Reviewer response [4. Proof reading service

With kind regards, 

Zelalem Desalegn Woldesonbet

Submitting author, 

Reviewers and editor comments 

Reviewer #1: The authors have sufficiently addressed all comments by the reviewer. The manuscript is now acceptable in its current form. Well done !

Reviewer #2: I am okay with all the corrections done by the author.

Editor Comments (if provided): 

1. Authors have adequately addressed the comments of reviewers. 

We are grateful for the scientific inputs given by the academic editor and reviewers as well.

2. Authors need to follow the journal guidelines in preparing the Reference section. 

Well taken and we used a Vancouver style as outlined in the ICMJE sample references ( National Library of Medicine) and considering previously published articles in Plos One journal to meet the journal requirements(Please refer line number 449 to 539).

3. Still, there is a need to correct some grammatical errors to improve readability and to meet the journal standard. 

Well taken. In addition to the professional proof reading service, the authors have thoroughly gone through the revised manuscript to improve the readability and meeting the journal standard.

Abstract: 

1. LINE 69: "The current study revealed that" (to replace shows)

Well taken. The stated phrase incorporated directly into the revised manuscript(Please refer line number 68).

Text

LINE 122: "Despite the extensive efforts made so far, accumulated evidence indicates that" (to remove shows) 

We appreciate the comment. As per the comment" shows: replaced with the word " indicates ( please refer line number 121).

LINE 160: "considering a KAP survey as a suitable format' 

Well taken. In the main document " is" replaced with " as" (please refer line number 150)

LINE 157: " A total of six government hospitals and four private hospitals were included in the study

The comment incorporated into the revised version (Please refer line number 156 to 158)

" LINE 162: "A written informed consent" 

Well taken and incorporated in the revised version ( Please refer line number 163).

LINE 168: "health professionals had knowledge and preparedness

Thank you. The comment incorporated into the revised version of the manuscript ( Please refer line number 169).

" LINE 169: "considering for a design effect of"

We appreciate. The comment considered in the document (Please refer line number 170 to 1710).

 LINE 218: "The study included 1,334 health professionals"

Thank you. We have considered the comment in the manuscript ( Please refer line number 171 to 172)

 LINE 229: "The finding showed that" 

Well taken. We have made the change ( Please refer line number 229).

LINE 253: "represented doctors

Well taken (Please refer line number 253).

" LINE 302: please correct as follows- "that there were reported deaths of over 3,300 (20%) HCWs [22]." 

Yes, we accept the way you have constructed the description (Please refer line number 298 to 299).

LINE 304: "HCWs were worried" 

We accepted the comment. Kindly edited in the revised version of the manuscript (Please refer line number 300)

LINES 308-309: Please correct as follows- "At this time of crisis" rather than "At a time like this".

Well taken and incorporated into the document (Please refer line number 306).

 LINE 319: "Our findings demonstrated that" 

Well taken. The comment incorporated into the revised version (Please refer line number 316).

LINE 417: "which were somehow" 

We are grateful. The change considered in the revised manuscript (Please refer line number 414).

LINE 426: "there should be a regulation"

Well taken ( Please refer line number 423 to 424).

 LINE 432: "participants are recruited to the study" 

Well taken with the respective change in the document (Please refer line number 429).

LINE 433: "where people had plenty of access"

We appreciate the comment and considered in the document (Please refer line number 430)

 LINE 434: "consequently, the findings only reflected a sneak peak" 

Well taken and indicated in the manuscript (Please refer line number 431 to 432)

LINE 438: "This study concluded that"

Well taken and indicated in the manuscript ( Please refer line number 435).

---

## [Editor Report · Decision Letter 3]

3 Dec 2020

Evaluation of COVID-19 related knowledge and preparedness in health professionals at selected health facilities in a resource-limited setting in Addis Ababa, Ethiopia

PONE-D-20-19732R3

Dear Dr. Desalegn,

We’re pleased to inform you that your manuscript has been judged scientifically suitable for publication and will be formally accepted for publication once it meets all outstanding technical requirements.

Kind regards,

Khin Thet Wai, MBBS, MPH, MA (Population & Family Planning Resear

Academic Editor

PLOS ONE
---

## [Editor Report · Acceptance letter]

10 Dec 2020

PONE-D-20-19732R3 

Evaluation of COVID-19 related knowledge and preparedness in health professionals at selected health facilities in a resource-limited setting in Addis Ababa, Ethiopia  

Dear Dr. Desalegn:

I'm pleased to inform you that your manuscript has been deemed suitable for publication in PLOS ONE. Congratulations! Your manuscript is now with our production department. 

Kind regards, 

on behalf of

Dr. Khin Thet Wai 

Academic Editor

PLOS ONE